# REM: Evaluating LLM Embodied Spatial Reasoning through Multi-Frame Trajectories*

**Jacob Thompson, Emiliano Garcia-Lopez & Yonatan Bisk**
Department of Computer Science
Carnegie Mellon University
Pittsburgh, PA 15213, USA
{jbthomps,egarcial,ybisk}@andrew.cmu.edu

## Abstract

Humans build viewpoint-independent cognitive maps through navigation, enabling intuitive reasoning about object permanence and spatial relations. We argue that multimodal large language models (MLLMs), despite extensive video training, lack this fundamental spatial reasoning capability, a critical limitation for embodied applications. To demonstrate these limitations and drive research, we introduce REM (Reasoning over Embodied Multi-Frame Trajectories), a benchmark using controllable 3D environments for long-horizon embodied spatial reasoning. REM systematically evaluates key aspects like object permanence/distinction, spatial relationships, and numerical tracking across dynamic embodied viewpoints. Our evaluation shows that the best-performing current models exhibit promising overall performance, but become increasingly unreliable at even moderate complexity levels easily handled by humans. These findings highlight challenges MLLMs face in developing robust spatial representations from sequential visual input. Consequently, REM provides targeted metrics and diagnostics to foster improved spatial understanding in future models.

## 1 Introduction

When navigating a familiar campus, park, or neighborhood, we rarely consult external maps or need explicit directions. Instead, we rely on sophisticated mental representations built through experience. These three-dimensional cognitive maps (Tolman, 1948; O'Keefe & Nadel, 1978) function as accessible spatial databases that persist independently of our current viewpoint. Our mental maps encode precise spatial relationships ("standing near the main gate, the statue is to the left of the library") and allow us to plan optimal routes through areas currently out of sight ("cutting through the science building will save time"). This remarkable ability to maintain and interrogate comprehensive spatial representations across changing viewpoints even enables us to answer difficult questions like "How many blue recycling bins are on campus?" through explicit mental enumeration, drawing on our persistent cognitive model rather than immediate visual perception. This fundamental spatial reasoning capability, which humans develop naturally, represents a critical challenge for current AI systems attempting to understand embodied environments.

Multimodal Large Language Models (MLLMs) have emerged as powerful tools, demonstrating impressive capabilities on static image and video understanding tasks. Their ability to process and reason about visual and textual information makes them promising candidates for robotics and embodied AI applications. However, success in complex, dynamic environments requires more than recognizing objects or actions in isolated frames; it demands robust embodied spatial reasoning. This entails building and maintaining persistent internal models of 3D scenes—akin to human cognitive maps—that accurately encode spatial relationships and object identities even as they move in and out of view during navigation.

---

*Dataset and Code: github.com/EmilianoGarciaLopez/REM

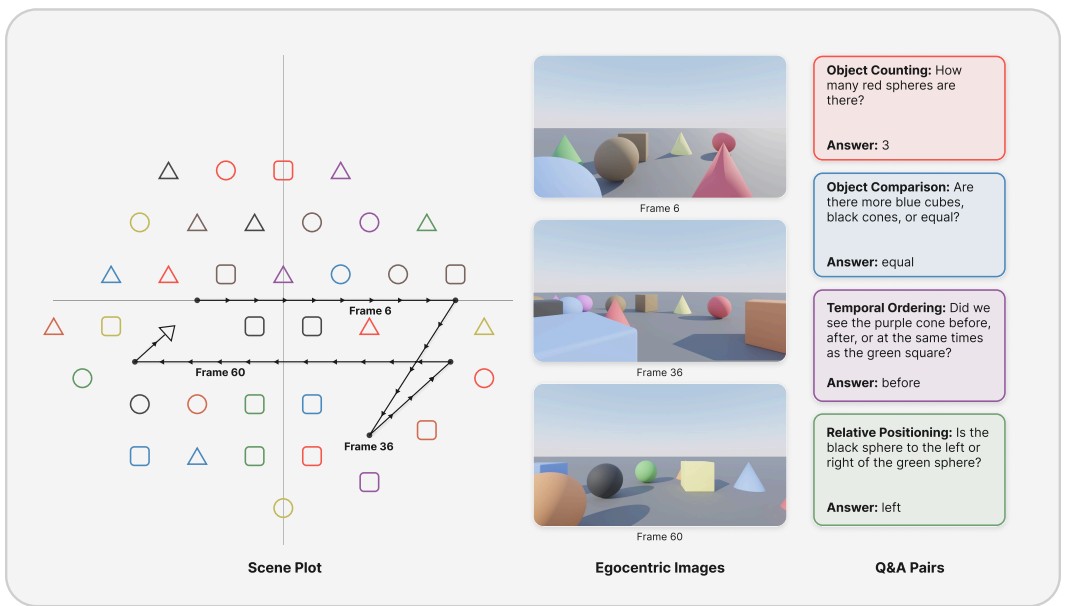

Figure 1: **REM at a glance**. Left: top-down plot showing object distribution and camera trajectory. Center: egocentric views from selected frames, simulating an agent's perception during navigation. Right: example question-answer pairs that test different aspects of spatial reasoning: counting, comparison, temporal ordering, and left/right relative positioning.

To investigate the extent to which current MLLMs possess these capabilities, we introduce **REM** (**Reasoning over Embodied Multi-Frame Trajectories**). REM is a novel benchmark specifically designed to evaluate embodied spatial reasoning using multi-frame visual sequences simulating egocentric navigation within synthetic 3D environments of controlled complexity. Unlike benchmarks focusing solely on static images (e.g., Goyal et al., 2017; Hudson & Manning, 2019; Liu et al., 2024a) or complex, uncontrolled video scenarios (e.g., Mangalam et al., 2023; Li et al., 2024a), REM allows for systematic probing of fundamental spatial understanding across changing viewpoints. It assesses critical capacities including object permanence (tested via numerical counting across frames), spatial relationships (tracking left/right positioning), temporal ordering (understanding appearance sequence), and the ability to integrate visual context with movement history for object identification.

Our evaluation using REM reveals significant shortcomings in state-of-the-art MLLMs. While newer models show impressive capability in reasoning over simpler scenes, as shown in our 'baseline' dataset, performance degrades rapidly with increased complexity. Furthermore, models fundamentally struggle with object permanence and distinction, particularly under challenging viewpoint shifts, as demonstrated in our 'Full Rotation' experiments where performance typically collapses. These findings underscore the specific, deep challenges MLLMs face in developing robust spatial representations from sequential visual input, highlighting the benchmark's utility in driving targeted improvements for embodied AI.

## 2 REM: Reasoning over Embodied Multi-Frame Trajectories

Summarized in Table 1, we introduce REM, a set of three Blender-generated egocentric datasets designed to quantitatively evaluate multimodal large language models' abilities in embodied visuospatial reasoning, specifically focusing on object permanence and distinction, spatial relationships, and accuracy in counting and numerical object comparisons. Each dataset consists of trajectories composed of sequential egocentric image frames. Crucially, the discrete camera action taken between consecutive frames (e.g., 'move forward 1m', 'rotate right 15°') is explicitly provided alongside the image sequence as input to the model,

simulating an agent aware of its own movements. These trajectories occur within simple synthetic environments containing objects of visually distinct shapes (`cuboids`, `spheres`, `cones`) and colors (`brown`, `yellow`, `red`, `green`, `blue`, `purple`, `black`, `orange`).

These environments are intentionally made to be simple, with easily distinguishable object shapes and colors, as REM aims to diagnostically pinpoint failures and performance scaling behavior in visuospatial reasoning and memory, not object detection or visual distinction.

REM consists of three datasets—**Baseline**, **Single Frame**, and **Full Rotation**. The **Baseline** dataset benchmarks general capabilities across carefully varied task complexities; **Single Frame** isolates single-image counting ability as a control; and **Full Rotation** specifically probes object permanence/distinction and contextual reasoning under challenging viewpoint changes.

## 2.1  Baseline dataset

We introduce our baseline dataset as the foundation of REM, designed to comprehensively evaluate MLLM performance across four visuospatial reasoning tasks. It contains nearly 50,000 question-answer pairs distributed across over 3,000 embodied image-action trajectories. These trajectories systematically vary three key dimensions:

- **Trajectory Length**: Sequences contain either 2, 4, 8, 16, 32, or 64 image frames with associated actions, to evaluate how performance scales with increasing context length and scene views.
- **Scene Congestion**: Environments contain varying object densities, with either 8, 16, 24, 36, or 48 total objects to test performance under different visual complexity conditions.
- **Object Duplication**: Scenes range from all-unique objects to mostly duplicated objects (as few as two distinct types), testing identical object individuation and tracking.

Trajectory lengths are distributed uniformly, while trajectory object densities and total duplicate counts can be found in Figure 10 of the Appendix. The baseline dataset includes four primary question categories, meant to probe different aspects of visuospatial reasoning:

- **Object Counting**: "How many blue objects are there?"
- **Comparison**: "Are there more red cones, spheres, or equal?"
- **Relative Positioning**: "Is the green sphere to the left or right of the blue cuboid?"[1]
- **Temporal Ordering**: "Did we see the purple cone before, after, or same time as the orange cuboid?"

Each trajectory is automatically generated to include a variety of movement patterns, with randomized sequences of both forward movements and rotations. We vary both the number of consecutive forward movements before rotating (randomly left or right) and how many consecutive 15° rotations occur in a single direction, ranging from single 15° turns to complete 180° (or greater) viewpoint changes, as permitted by the remaining trajectory length. This design ensures that for longer trajectories, models cannot rely solely on tracking objects through small, incremental camera changes, but must maintain object representations across significant viewpoint shifts where objects completely leave and re-enter the field of view. An example length-4 trajectory is depicted in Figure 2.

### 2.1.1  Mini-Baseline dataset for Human Comparison

We also sample a representative Mini-Baseline subset (18 trajectories; 154 QA pairs) from the larger dataset for a human performance comparison. Participants were given unlimited time, provided a physical scratchpad, and used an interactive interface that allowed them to use arrow keys to freely navigate images in the trajectory, similar to a reasoning model

---

[1]Importantly, this question is only asked in trajectories where objects maintain their left/right positioning throughout all appearances in the trajectory.

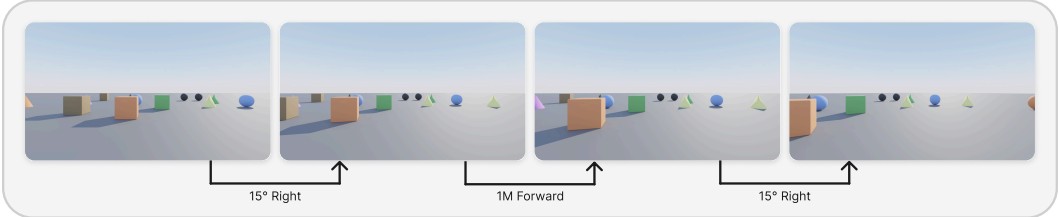

Figure 2: **Example length-4 trajectory from the baseline dataset.** Models receive the sequence of egocentric visual frames and the corresponding discrete actions ('15° Right', '1m Forward', '15° Right') taken between frames. Evaluating performance on such sequences tests the model's ability to integrate visual perception with known movement for spatial reasoning across changing viewpoints.

repeatedly accessing its context while producing reasoning tokens. Mini-Baseline trajectory statistics can be found in Figure 11 of the Appendix.

## 2.2 Single Frame dataset

The Single Frame dataset, consisting of 350 trajectories and approximately 1,300 QA pairs, isolates visual counting ability by eliminating frame-to-frame tracking requirements. This control condition helps disentangle whether counting errors in sequential settings originate from fundamental perceptual limitations, from failures in maintaining object identity across frames, or a mixture of both. Unlike the baseline dataset, we include only object counting questions, providing comparative baselines for our multi-frame evaluations. Dataset object and total duplicate statistics can be found in Figure 9 of the Appendix.

## 2.3 Full Rotation dataset

The Full Rotation dataset, containing 100 trajectories and approximately 2,400 QA pairs, challenges models with a trajectory consisting of a 360° rotation in a cluttered scene. At the 180° mark, the scene deliberately mirrors the initial 0° view: while the visual arrangement appears identical, most objects are different, with 1-2 target objects intentionally duplicated. Importantly, the rotating camera maintains a continuous stream of object views, testing whether MLLMs distinguish identities through integrated contextual and movement cues or use simpler heuristics like spatially invariant attention aggregation. An example is illustrated in Figure 3.

## 2.4 Automated Question-Answer (QA) generation and verification

In REM, each trajectory is automatically generated with Blender-provided per-frame ground truth object annotations, specifying color-shape identities and left-to-right spatial positioning. We then construct detailed data dictionaries recording object counts by color, shape, and color-shape combinations, along with the specific frames where each object appears.

We automatically generate question-answer pairs using a template system for each question type. For left/right relative positioning and temporal ordering questions, we randomly select two color-shape entities (e.g., "red cube," "blue sphere") and consult our ground truth annotations to determine correct answers. For counting and numerical comparison questions, we randomly select entities (which may be specific color-shape combinations, colors, or shapes) and reference our ground truth counts.

For evaluation, we employ a keyword-based verification approach that identifies terms uniquely associated with correct answers. For instance, in counting questions, we extract the first numerical value. For positioning questions, we search for a single directional indicator ("left"/"right") - if both are provided, the answer is considered wrong per our system prompt. For temporal questions, we look for a single sequence markers ("before"/"after").

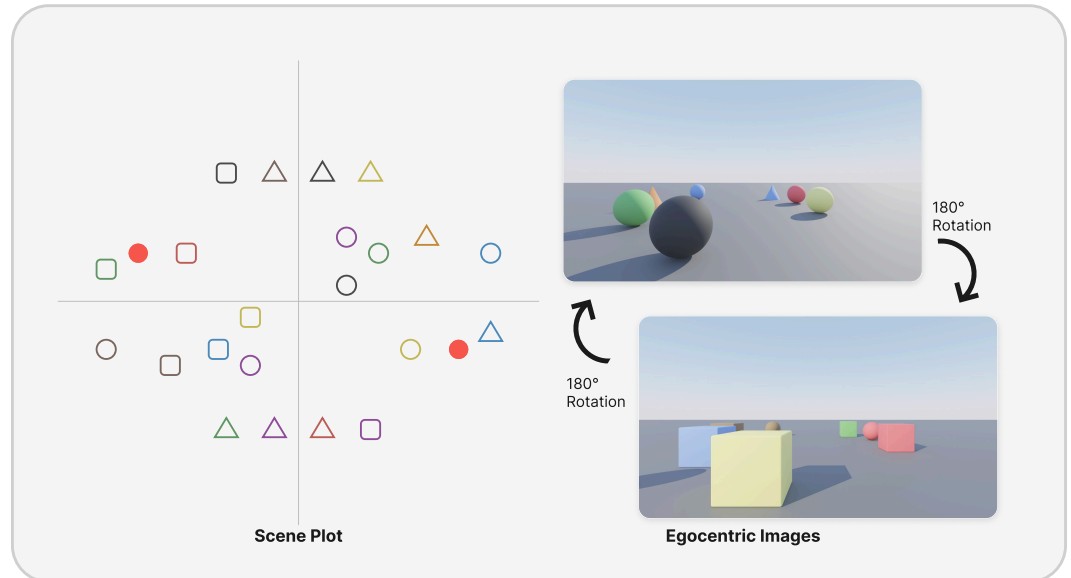

Figure 3: **Full Rotation dataset example scene**: (left) top-down scene layout showing object positions, with the camera at the origin, (top-right) view at $0°$, and (bottom-right) view after $180°$ rotation. Note the red sphere is intentionally duplicated between views, while other objects occupy identical spatial positions but are visually distinct entities. Peripheral objects at the top and bottom of the scene layout maintain visual continuity during camera transitions, preventing empty frames with movement ambiguity.

To ensure maximum fairness to models, we exclude any question where our verification system cannot identify unambiguous correctness markers (note this is exceedingly rare). A detailed diagram depicting the dataset generation and verification pipeline, as well as the prompting format, are provided in Figures 12 and 13 of the Appendix.

| Property | Baseline | Single Frame | Full Rotation |
|---|---|---|---|
| Num. Trajectories | 3,119 (18) | 350 | 100 |
| Total QA Pairs | 47,019 (154) | 1,289 | 2,424 |
| Trajectory Length(s) | $2, 4, 8, 16, 32, 64$ $(4, 8, 16, 32)$ | 1 | 24 |
| Object Count | 8-48 | $24 - 55$ | 24 |
| Duplicate Count | $0 - 46$ | $0 - 20$ | $1 - 2$ |
| Purpose | General Capabilities | Single-frame Counting | Object Distinction |

Table 1: Comparative summary of the REM component datasets (Baseline, Single Frame, Full Rotation), detailing differences in dataset size, trajectory lengths, scene complexity, and evaluation purposes. Mini-Baseline dataset for human evaluation shown in parentheses.

## 3  Dataset evaluation and results

### 3.1  Baseline results overview

We evaluate seven multimodal models: OpenAI o3 and GPT-4o, three from the Gemini family (2.5-Pro, 1.5-Pro, 1.5-Flash), Nova-Lite-v1, and Llama-3.2-11B. The new SOTA reasoning models (o3, 2.5-Pro) significantly outperform the older non-reasoning models, with the lightweight proprietary Nova-Lite-v1 and open-source Llama-3.2-11B performing only slightly above chance. Note Gemini-2.5-Pro is only tested on the Mini-Baseline.

o3 performs the best on the full Baseline with an impressive overall performance of 80.0%, but with the exception of left/right relation questions, performs noticeably worse than the near-perfect human baseline. Notably, o3 comparatively struggles on counting and numerical comparison tasks, which require aggregating information from multiple objects across multiple images while respecting object permanence and distinction. A more thorough analysis of individual question performance and scaling laws follows in Section 3.2.

| Question Metrics | Overall | Num. Comparison | Left/Right Rel. | Temp. Ord. | Counting |
|---|---|---|---|---|---|
| Full Count | 47,019 | 15,580 | 1,576 | 14,304 | 15,559 |
| Mini Count | 154 | 39 | 38 | 37 | 40 |
| Random Chance | – | 33.3 | 50.0 | 33.3 | – |
| **Models** | **Full Baseline (Accuracy %)** | | | | |
| o3 | 80.0 | 78.3 | 92.3 | 88.4 | 60.9 (35.7) |
| GPT-4o | **61.7** | 57.9 | 61.4 | **73.0** | **54.3 (24.4)** |
| Gemini-1.5-Pro | 59.6 | 54.3 | **67.7** | 69.0 | 47.4 (14.1) |
| Gemini-1.5-Flash | 58.2 | **59.5** | 59.5 | 65.8 | 47.8 (13.8) |
| Nova-Lite-v1 | 38.7 | 38.1 | 51.3 | 43.4 | 21.9 (7.3) |
| Llama-3.2-11B | 37.3 | 31.6 | 51.5 | 45.7 | 20.2 (7.0) |
| **Models** | **Mini Baseline (Accuracy %)** | | | | |
| Human Average | **97.8** | **96.3** | 99.3 | **98.2** | **97.5 (97.7)** |
| o3 | 80.1 | 64.1 | **100.0** | 83.8 | 72.5 (59.1) |
| Gemini-2.5-Pro | 79.4 | 74.0 | 78.9 | 94.6 | 70.0 (54.5) |
| Gemini-1.5-Pro | **58.4** | 51.3 | **71.1** | 51.4 | **60.0 (36.4)** |
| GPT-4o | 57.1 | **56.0** | 57.9 | **59.5** | 55.0 (31.8) |

Table 2: Top: dataset metrics. Middle: evaluation on the full Baseline. Bottom: evaluation on the mini baseline sample. Reasoning models separated from non-reasoning models. Overall performance shown in descending order. Counting question performance with ground truth $\geq 2$ shown in parentheses.

## 3.2 Baseline performance breakdown

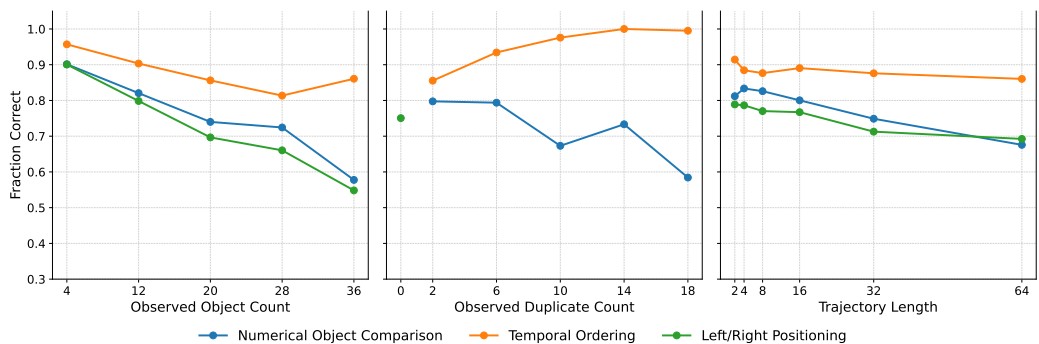

Figure 4: **o3 QA performance across three scaling factors: (a) observed object count, (b) observed duplicate count, and (c) trajectory length.** Curves show average correctness for Numerical Object Comparison, Temporal Ordering, and Left/Right Positioning tasks.

Figure 4 demonstrates basic scaling laws of non-numerical questions with overall object count, total number of observed duplicates, and trajectory length. For simplicity, we limit our analysis to o3, although similar scaling laws are observed with other models.

**Object Comparison.** We observe greatly decreasing performance with overall scene congestion (both observed object count and duplicate count), with the most complex scenes (36

viewed objects) dropping below 60% overall accuracy. Similarly, a moderate decrease in performance is also observed with increasing trajectory length beyond 4 images. In both cases, this is likely due to decreased signal-to-noise ratio (occlusion, scene diversity) of the two objects being compared, and a greater need to sift through the scene for the correct objects.

Additionally, as shown in Figure 5, we notice a significant decrease in model performance as targeted object counts become more similar, with 0 difference in target object quantities resulting in only 66% accuracy. Qualitatively comparable performance drop-off with GPT-4o and Gemini suggests models are using a kind of "fuzzy counting" heuristic that begins to fail with decreasing difference in target object quantities.

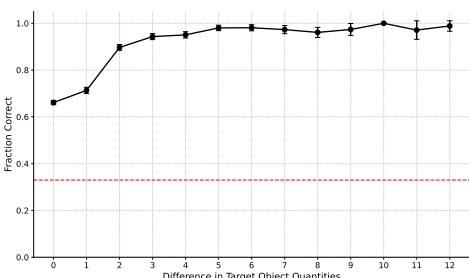
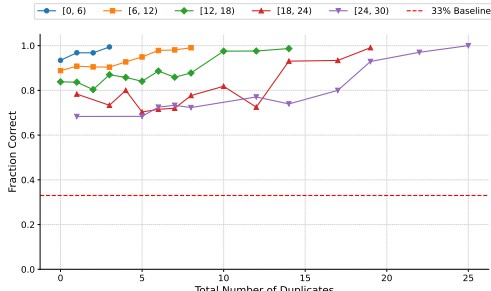

Figure 5: **o3 numerical object comparison count accuracy vs. the difference in target object counts.** Includes 95% confidence interval. Random 33% baseline provided in red.

Figure 6: **o3 temporal ordering question performance as a function of observed duplicate and total object counts.** Different colored lines bin respective object counts, with random 33% baseline provided in red.

**Temporal Ordering.** We initially observe a strong reduction in performance with increasing scene congestion (object count), but unlike the other question types, not with trajectory length. Interestingly, we also see a strong positive correlation with maximum duplicate counts, with the highest counts reaching saturation. The explicit relationship between duplicate count and overall object count is more directly observed in Figure 6, which shows that even in the most congested scenes, increasing duplicate count can lead to performance saturation. This is likely due to increasing duplicate count resulting in order-preserving questions being more likely to target the duplicates, which have an increased signal-to-noise ratio (less attention competition) in the scene.

**Left/Right Orientation.** In Figure 4, we notice left/right questions scaling strongly with scene congestion and moderately with trajectory length. An intuitive potential explanation for the scene congestion scaling is provided by increased occlusion, object attention competition, and more non-target objects appearing between the targets. Performance degradation with trajectory length is similarly intuitively explained by increasing scene complexity.

**Counting Questions.** Counting question performance is predominantly dependent on target object ground truth count. As shown in Figure 7, o3 progressively undercounts objects as ground truth increases. In fact, counting questions with target object class ground truth $\geq 2$ have an overall accuracy of only 35.7%.

Additionally, Figure 7 demonstrates that, independent of ground truth count, increasing trajectory length does not seem to consistently affect undercounting. This is perhaps counterintuitive, as longer trajectories include more diverse viewpoints of persistent objects, which would naively result in relative overcounting.

Furthermore, we observe in Figure 8 that while the ground truth number of target objects seen throughout the entirety of a trajectory is typically greater than the number observed by models in any single frame, the amount predicted by o3 closely tracks the latter value. Based on these observations alone, it is not clear whether counting is considering overall

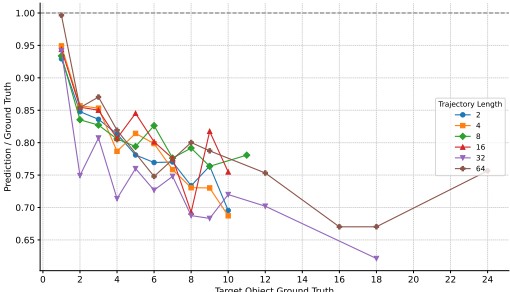 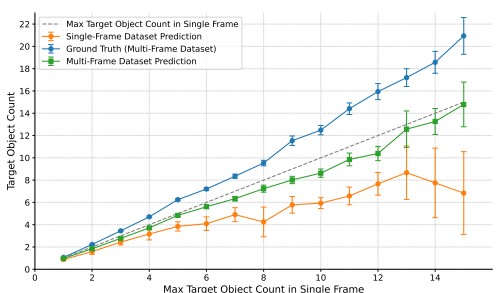

Figure 7: **o3 average undercounting fraction as a function of target object ground truth count.** Different lines indicate different trajectory lengths.

Figure 8: **o3 counting predictions vs. maximum number of target objects in a single frame.** Includes both single-frame and multi-frame trajectory ground truths, both with 95% confidence interval.

trajectory object counts (requiring some notion of object permanence and continuity) or if the model is simply selecting some subset (potentially the maximum object frame) of the trajectory, and returning a value correlated with object count in that subset.

It is also notable that while absolute counting performance is increasingly poor beyond ground truth counts $0 - 1$, o3 does retain a strong notion of *more* vs *less* objects in a trajectory, explaining why object comparison results are superior to absolute counting in higher duplicate count (and thus higher target object count) settings.

### 3.3 Single frame counting results

Similar to baseline counting, o3 exhibits systematic undercounting behavior in single-frame scenarios while maintaining a strong correlation with true object counts. This pattern again indicates that although absolute counting accuracy is limited, models retain a meaningful notion of relative quantity (Figure 8).

By averaging the fraction of undercounting observed with ground truth $\geq 2$ in Figure 8, we see that o3 undercounts with an average prediction-to-actual ratio of 0.65 (compared to 0.73 in multi-frame with same average ground truth), and progressively undercounts worse with increasing ground-truth values. Notably, we never observe overcounting. This improved undercounting ratio for multi-frame trajectories vs. single-frame trajectories (approximately 13% higher predictions), suggests that o3 is doing some kind of limited object count aggregation across frames.

### 3.4 Full Rotation results

Table 3 shows that the best-performing model, OpenAI o3, consistently undercounts in both single and double duplicated object scenarios, with a slightly less severe bias when only a single object is duplicated. This result is intuitive: the single-duplicated case features more distinct visual context after the 180° rotation, making the duplicated object scene easier to identify as different than the initial view. However, the dramatic undercounting when the ground truth is two (86% and 90% failure rates for single and double duplication, respectively) reveals a critical failure in integrating the complete trajectory. o3 fails to use the explicit motion cues and clear object differences to understand that visually similar configurations at 0° and 180° represent distinct sets of objects. Similar failure rates are seen for all other tested models, including GPT-4o and the Gemini-1.5 family.

| Ground Truth | Single Duplicated Object | | | Two Duplicated Objects | | |
|:---:|:---:|:---:|:---:|:---:|:---:|:---:|
| | Undercount | Correct | Overcount | Undercount | Correct | Overcount |
| 0 | 0.00 | 1.00 | 0.00 | 0.00 | 0.95 | 0.05 |
| 1 | 0.09 | 0.91 | 0.01 | 0.06 | 0.93 | 0.01 |
| 2 | 0.86 | 0.14 | 0.00 | 0.90 | 0.10 | 0.00 |

Table 3: **Impact of intentional object duplication on counting accuracy in the Full Rotation dataset with OpenAI o3.** Counting questions target objects with ground truth counts of either 0, 1, or 2.

## 4 Related work

**Introducing MLLMs.** Multimodal Large Language Models (MLLMs) typically combine pretrained vision encoders (Radford et al., 2021), (Dosovitskiy et al., 2021) with large language models (LLMs) (Brown et al., 2020), (Chowdhery et al., 2022), (OpenAI et al., 2024), (Touvron et al., 2023), (Vaswani et al., 2023). This integration enables them to process and reason about both visual and textual information, demonstrating strong capabilities in tasks such as visual question answering, image captioning, and generating descriptions grounded in visual input (Yin et al., 2024). The potential of these models extends to applications requiring interaction with a physical world, including robotics and embodied AI, where understanding dynamic scenes and spatial contexts is essential (Driess et al., 2023), (Brohan et al., 2023), (Wang et al., 2023), (Liu et al., 2025).

**MLLM Benchmarks for Static Images.** A substantial body of work focuses on evaluating MLLMs using single static images. Benchmarks utilizing synthetic scenes like CLEVR (Johnson et al., 2017) assess compositional reasoning and spatial relationships in controlled settings, while those grounded in real-world images, such as VQA v2 (Goyal et al., 2017), GQA (Hudson & Manning, 2019), and NLVR2 (Suhr et al., 2019), evaluate understanding and reasoning based on a single naturalistic visual input. Furthermore, comprehensive evaluation suites including MME (Fu et al., 2024a) and MMBench (Liu et al., 2024a) provide broader assessments across diverse perception and cognition tasks within a single-image context. While invaluable for measuring progress in interpreting complex visual scenes, these static evaluations highlight the need for distinct methodologies capable of addressing the challenges posed by temporal dynamics, viewpoint changes, and the continuous perception-action loop characteristic of embodied agents.

**MLLM Benchmarks for Video and Embodied Spatial Reasoning.** To evaluate performance on dynamic inputs, numerous video-based benchmarks have been proposed, targeting various aspects of understanding. Early benchmarks often focused on temporal event comprehension, action recognition, or causal reasoning within videos (Jang et al., 2017; Lei et al., 2019; Yi et al., 2020; Girdhar & Ramanan, 2020), with diagnostic datasets further probing specific temporal concepts (Li et al., 2024b; Liu et al., 2024b). Subsequently, comprehensive benchmarks emerged to assess broader video understanding across diverse content and tasks (Li et al., 2021; Fu et al., 2024b; Li et al., 2024a; Ning et al., 2023; Fang et al., 2024). More recently, reflecting the push towards embodied AI, evaluations have increasingly incorporated embodied perspectives, revealing challenges in areas like long-form egocentric understanding (Mangalam et al., 2023), embodied question answering (Majumdar et al., 2024), and spatial reasoning (including explicit counting) using realistic trajectories from 3D scans (Yang et al., 2024). While these benchmarks effectively identify performance gaps on complex tasks, REM complements them by utilizing a controlled synthetic environment specifically designed to enable detailed analysis of how fundamental factors influence core spatial reasoning performance. By precisely manipulating trajectory length, scene clutter, and object duplication, REM allows for a systematic study of their impact on object permanence (critically tested via counting across views), spatial relationship tracking under viewpoint shifts, and numerical consistency during simulated movement. This focused approach provides targeted diagnostics on where models might fail, offering insights beyond overall performance metrics on complex scenarios.

## 5 Discussion

We introduced REM, a benchmark leveraging controlled synthetic environments to probe MLLM embodied spatial reasoning across sequential egocentric viewpoints. Our analysis suggests a core limitation: **MLLMs struggle to build and maintain a stable internal world model that integrates visual perception with explicit discrete egomotion to track distinct object instances over time and viewpoint changes.** This fundamental deficit manifests in the models' handling of object identity and quantity across dynamic scenes:

The poor world modeling results in **deficient spatiotemporal context integration**, meaning models fail to correctly update their internal scene representation using the explicitly provided actions alongside unfolding visual context (Section 3.4). This is best shown in the 'Full Rotation' experiment (Table 3), where models cannot leverage known motion and visual change to disambiguate visually similar scenes. They incorrectly merge distinct object instances seen before and after the 180° turn, demonstrating a failure to ground perception in movement and visual context to form a viewpoint-invariant understanding.

This difficulty in individuating objects using spatiotemporal context possibly contributes to the observed **poor numerical grounding** (Sections 3.2, 3.3). When distinct instances are erroneously merged due to context integration or object representation failures, systematic undercounting can naturally follow, extending observed under-counting biases in single-frame perception to multi-frame trajectories.

The impact of this core limitation additionally varies across other specific tasks. 'Counting' directly probes permanence (degrading with ground truth count, Figure 7). 'Numerical Comparison' reflects noisy object quantity representations (failing on small differences, Figure 5). 'Left/Right' reasoning shows decaying relational tracking with distance and clutter (Figure 4). 'Temporal Ordering' highlights sensitivity to *distinct* instance tracking versus overall scene complexity (improving with duplicates, degrading with congestion, Figure 6). Unlike static or uncontrolled video benchmarks, REM's systematic variation isolates these specific failures in dynamic spatial reasoning. For example, REM shows that by simply increasing scene congestion from the Baseline dataset average to 36 viewed objects, and only including counting questions with ground truth $> 1$, overall SOTA model performance drops from an impressive 80% to under 60%.

This inability to form integrated world models (cognitive maps) is a primary bottleneck for deploying MLLMs in embodied AI. Future work must prioritize architectures and training paradigms that explicitly foster robust object permanence and the integration of spatiotemporal context, using targeted benchmarks like REM to measure progress towards spatially grounded intelligence.

## Acknowledgments

This work was partially supported by funding from Lockheed Martin Corporation, the Distributed Collaborative Intelligent Systems and Technology (DCIST) Collaborative Research Alliance, and Fujitsu Limited.

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

# A    Dataset Statistics

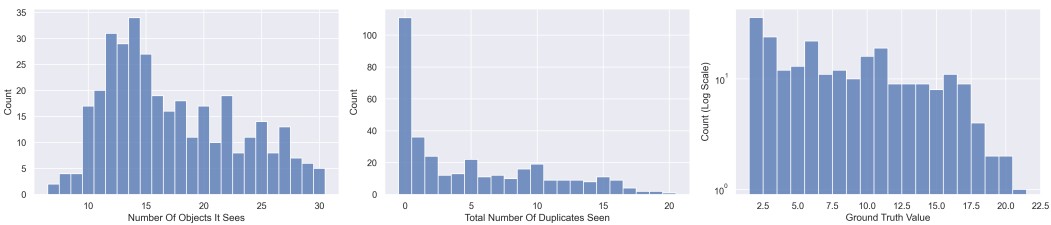

Figure 9: **Key Statistics for the Single Frame dataset.** Left to right: Objects seen per frame (mean 17.3, range 7-30); Duplicates seen (mean 5.01, range 0-20); Ground truth counts for counting questions ($\geq 2$, log scale), showing a long-tailed distribution relevant to undercounting analysis (see Figures 7 and 8). Note the log scale on the rightmost plot.

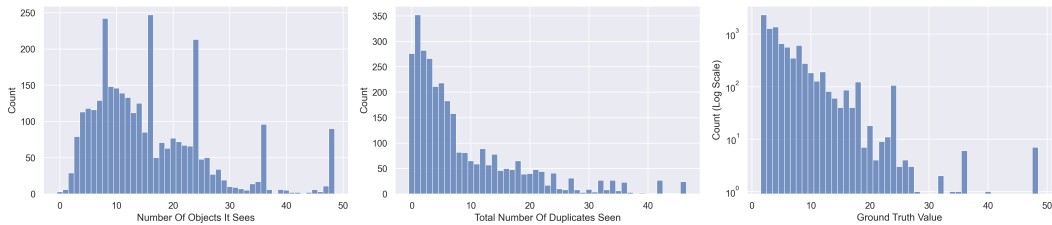

Figure 10: **Key Statistics for the Baseline dataset.** Left to right: Objects seen per trajectory; Duplicates seen per trajectory; Ground truth counts for number questions ($\geq 2$, log scale). Note the long-tailed distribution for counts, also relevant to undercounting analysis (see Figures 7 and 8). Note the log scale on the rightmost plot.

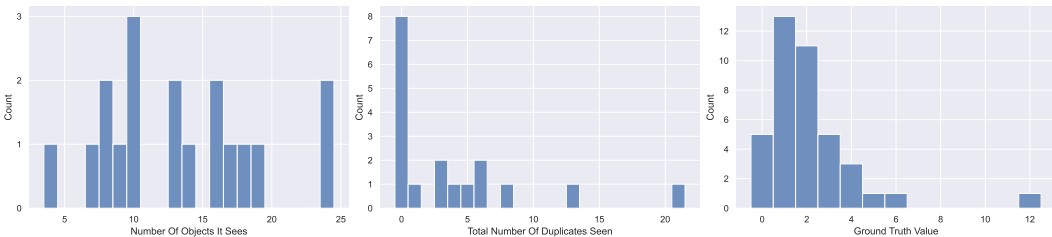

Figure 11: **Key Statistics for the Mini Baseline dataset.** Left to right: Objects seen per trajectory; Duplicates seen per trajectory; Ground truth counts for number questions. Note that the eight zero duplicate trajectories contain only left/right questions.

## B    Dataset generation and evaluation Details

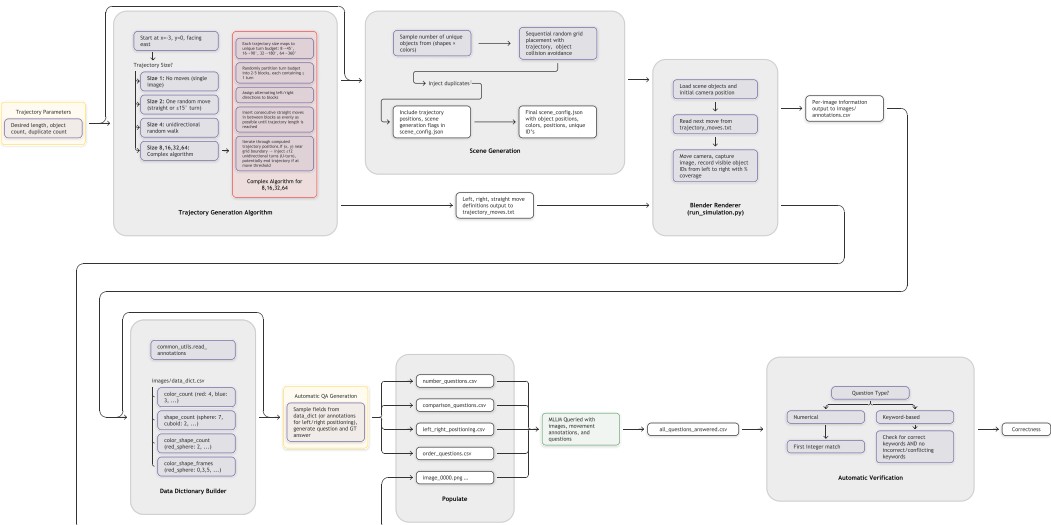

Figure 12: **Generation and evaluation pipeline for REM.** Starting from trajectory parameters (length, object count, duplicate rate), we synthesize a collision-free scene (sample shapes/colors and inject duplicates) and a discrete egomotion plan, then render the egocentric sequence in Blender while logging per-frame object IDs and pixel coverage. The annotations feed a data dictionary that aggregates counts and frame indices, from which we automatically instantiate QA templates (counting, comparison, left/right positioning, temporal ordering). Models are queried with the images, actions, and questions, and a rule-based verifier (numeric/keyword checks) scores answers for correctness.

---

### System Prompt

You are an agent walking through an environment in Blender. You will receive a series of images, each taken after taking an action in the environment (either moving straight or turning 15 degrees left/right). You will also receive a question that you must answer correctly after seeing all images. You will see objects with a shape and a color. The possible shapes include cuboid, cone, sphere. The possible colors include red, green, blue, yellow, purple, brown, black, orange. Please answer the question based on the set of images. Answer as concisely as possible, usually only a single word. If you're asked about a true/false question, answer with 'yes' or 'no' only. If it's a question where you're asked to compare the number of objects, respond only with whichever object there are more of, or equal, if there are the same number of objects If you're asked to count objects, answer only with the number (as a number, not in english) of objects you see. If you're asked whether you saw something before, after, or at the same time as another object, answer only with 'before', 'after', or 'same time' only. If the first time you see an object is in an image before another object, it comes before (and the other comes after). If two objects appear in the same frame together for their first viewing, its same time

---

Figure 13: **System prompt and message format.** Each model query contains two messages: the fixed system prompt (above), then a single user message whose content is, in order: the question text, the movement annotations, and all trajectory images, each resized to $960 \times 640$.

