# OpenReview forum: "REM: Evaluating LLM Embodied Spatial Reasoning through Multi-Frame Trajectories"
_colmweb.org/COLM/2025/Conference — COLM 2025_

### Official Review · Reviewer_gAU9 · 2025-05-10

**Rating:** 5
**Confidence:** 4
**Ethics Flag:** 1

**Summary:**

This paper introduces REM, a new benchmark to evaluate long-horizon, multi-frame spatial reasoning. The benchmark comprises synthetic 3D scenes generated in Blender using primitive objects like cubes and cylinders. Random camera trajectories are simulated to capture the environment. Four types of question-answer pairs are generated using a template-based system, including object counting, numerical comparisons, relative positioning, and temporal ordering. The dataset includes three variants: (1) a baseline for general spatial reasoning, (2) a single-frame version, and (3) a multi-frame 360° rotation from a fixed location. Several state-of-the-art MLLMs are evaluated on this benchmark. Results show that current MLLMs struggle to build a stable internal world model that integrates spatiotemporal context.

=== Update after author response ===

The rebuttal addressed some of my concerns including the benchmark construction and evaluation. However, I still have concerns regarding the unrealism of the benchmark and the lack of insights into the methodology. I increased my rating from 4 to 5.

**Reasons To Accept:**

1. The paper presents a controllable benchmark for testing spatial reasoning capabilities in MLLMs.
2. The analysis of GPT-4o’s performance across different question types and dataset variants is comprehensive.

**Reasons To Reject:**

1. The benchmark relies on overly simplified, synthetic environments, which limit its realism and applicability to real-world scenarios. No human evaluations are conducted on the benchmark.

2. There are concerns about benchmark construction and evaluation:
- It is unclear whether the randomly sampled frames fully capture the scene to allow meaningful reasoning.
- The keyword-based answer evaluation may be flawed. For example, if a model outputs both "left" and "right", it might be incorrectly considered fully correct.

3. The usage of MLLMs lacks clarity.
- Prompting strategies are not well-documented or explored.
- The potential for leveraging spatial depth or camera poses is not discussed.
- No method is proposed to improve model performance; the paper is purely evaluative.

---

> ### Author Response · Authors · 2025-06-02
>
> **If you have time, please read the general comment before reading your specific response, as we feel it builds on it.**
>
> Thank you for your comprehensive analysis. Below we address your specific concerns:
>
> 1. **Evaluation methodology.**
>    Your “left and right” example is important—our verification correctly marks such answers as incorrect. We use token‑subset checking: if the ground truth is “left,” the model’s output must contain “left” **and not** “right.” Any ambiguous response fails (rare with our prompting). For counting, we extract a single integer via regex; multiple or missing integers fail. This strict evaluation means only unambiguous correct answers pass. Full verification code will be provided, and the procedure will be documented in the appendix.
>
> 2. **Frame sampling and scene coverage.**
>    Frames are not randomly sampled; trajectories are. Each follows deterministic movements (15° rotations, 1 m forward steps). While coverage varies, questions are generated solely from what the camera sees and only when unambiguous. Details of the QA‑generation and verification pipeline will be added to the appendix.
>
> 3. **MLLM usage and prompting.**
>    Models receive (1) the full image sequence, (2) explicit actions between frames, and (3) clear instructions about the Blender environment, the task, and the required answer format. We use zero‑shot prompting with frame indexing (e.g., “Frame 0: [image], Action: rotate left 15°, Frame 1: [image] …”). Exact prompts and rationale will be documented in the appendix.
>
> 4. **Value of diagnostic benchmarks.**
>    Like GLUE in NLP, targeted evaluations drive progress by exposing weaknesses. REM’s systematic ablations show spatial‑reasoning failures scale predictably with scene factors (Figures 4‑7); the 80–89 % failure rate on Full Rotation highlights a fundamental MLLM limitation in object permanence.
>
> We appreciate your thorough review—your feedback strengthens REM’s contribution as a diagnostic tool.

---

> ### Author Response · Authors · 2025-06-06
>
> Thank you for your rebuttal feedback! We’re excited that CoLM has introduced this discussion week so we can respond to any specific concerns you still have about the methodology. If you have time, please feel free to include specific aspects you’d like clarified in your reply, and we’ll respond as quickly as possible.
>
> Regarding the unrealism concern: much like CLEVR’s “unrealistic” blocks proved key to advances in visual reasoning, we hope our controlled environments can do the same for embodied spatial reasoning by demonstrating fundamental limitations of LLM spatial memory.
>
> We look forward to your specific feedback on what methodological insights would be most valuable to include!

---

> > ### Author Response · Authors · 2025-06-09
> >
> > In addition to these comments, we’ve begun conducting a preliminary human baseline study (n=10 trajectories, 150 questions) with initial results showing humans achieve 95.0% overall accuracy while Gemini-1.5-Flash achieves 66% overall on the same subset.
> >
> > Per question type:
> > - **Numerical counting** (Human: 95.0%, Flash: 57.5%)
> > - **Temporal ordering** (Human: 100%, Flash: 81.1%)
> > - **Comparison** (Human: 87.2%, Flash: 61.5%)
> >
> > These substantial performance gaps suggest that our frame sequences provide sufficient context for accurate reasoning and that REM captures meaningful differences between human and MLLM embodied spatial reasoning capabilities.
> >
> > We appreciate reviewers highlighting the importance of human baselines — these results help contextualize model performance and demonstrate the benchmark’s diagnostic value. We will add complete results to Table 2 and include comprehensive methodology details in the appendix, including human dataset construction, interface design, and participant instructions.

---

### Official Review · Reviewer_2bj9 · 2025-05-12

**Rating:** 6
**Confidence:** 3
**Ethics Flag:** 1

**Summary:**

This paper introduces REM, a benchmark designed to evaluate spatial reasoning capabilities in multimodal large language models (MLLMs) within simulated 3D environments. REM offers a controllable and diagnostic Blender-based environment through tasks that test spatial relationships, temporal ordering, and numerical tracking across egocentric trajectories. The authors demonstrate that current state-of-the-art MLLMs struggle with REM, particularly under viewpoint shifts and increased scene complexity.

**Reasons To Accept:**

- The paper proposes a new benchmark to probe the spatial reasoning ability of VLMs with sequential visual inputs. REM challenges existing LLMs with controlled environment simulation, calibrated trajectories and multi-dimensional questions.
- The analysis reveals some interesting findings such as LLMs lack of an internal world model to be aware of its spatial positions or identify the same object across different frames.

**Reasons To Reject:**

- The template-based questions may introduce ambiguities in ground truth answers. It is not clear enough to me how the question generation process guarantees the correctness of the ground truth answer without ambiguity. For example, in Fig.1's relative position question, what if the center moves to a new point with a different camera angle where the black sphere appears to the right of the green sphere? How is that avoided in the trajectory sampling or question generation stage? In addition, is object occlusion considered when generating questions? How do the questions guarantee that the involved objects are visible in the frames?
- While REM comes with good motivation for probing LLM's awareness of its spatial positions, I am unsure if the provided frame inputs are sufficient to justify the argument. In the simulated environment with many similar/duplicated objects without natural background reference, even human beings may not be able to answer the questions correctly.
- REM is limited to rendered symbolic objects based on limited colors and shapes. It may not reflect LLMs true capabilities when placed in a more realistic environment.

---

> ### Author Response · Authors · 2025-06-02
>
> **If you have time, please read the general author comment as we feel your specific response builds on it.**
>
> Thank you for recognizing REM's value in revealing MLLMs' lack of internal world models.
> We appreciate your specific concerns:
>
> 1. **Ground‑truth ambiguity.**
>    Your example perfectly illustrates our robustness checks. If the black sphere ever appears right of the green sphere, that pair is filtered out—no question is generated. Concretely:
>    1. We project 3D positions to camera coordinates for *every* frame.
>    2. We track left/right consistency across the trajectory; if black is left of green in frames 1, 5, 8 but right in frame 12, the pair is rejected.
>    3. A render‑pass pixel count yields exact visibility percentages; only objects with > 0 % visibility are included.
>    Thus every generated question has exactly one correct answer. We will add visual diagrams of this process. Ground truth is derived from Blender’s scene graph, mirroring CLEVR’s methodology.
>
> 2. **Human performance without natural backgrounds.**
>    This setup is why we expect humans to excel—the task isolates spatial reasoning. Models are explicitly told (via prompts in the appendix) that objects are distinct Blender shapes on a ground plane and must be tracked through the trajectory. The Full‑Rotation experiment—where GPT‑4o fails 80–89 % of the time—would be trivial for humans who recognize that views at 0° and 180° contain different objects. A human baseline is forthcoming.
>
> 3. **Symbolic objects.**
>    Using maximally discriminable shapes is intentional and diagnostic. When models fail catastrophically even under ideal perceptual conditions, the deficit can only stem from spatial reasoning, revealing fundamental architectural limitations.
>
> Your insightful questions help us clarify why REM's controlled approach is essential for uncovering these core deficits.

---

> > ### Author Response · Authors · 2025-06-09
> >
> > In addition to our previous comment, we’ve begun conducting a preliminary human baseline study (n=10 trajectories, 150 questions) with initial results showing humans achieve 95.0% overall accuracy while Gemini-1.5-Flash achieves 66% overall on the same subset.
> >
> > Per question type:
> > - **Numerical counting** (Human: 95.0%, Flash: 57.5%)
> > - **Temporal ordering** (Human: 100%, Flash: 81.1%)
> > - **Comparison** (Human: 87.2%, Flash: 61.5%)
> >
> > These substantial performance gaps suggest that our frame sequences provide sufficient context for accurate reasoning and that REM captures meaningful differences between human and MLLM embodied spatial reasoning capabilities.
> >
> > We appreciate reviewers highlighting the importance of human baselines — these results help contextualize model performance and demonstrate the benchmark’s diagnostic value. We will add complete results to Table 2 and include comprehensive methodology details in the appendix, including human dataset construction, interface design, and participant instructions.

---

### Official Review · Reviewer_eEuK · 2025-05-14

**Rating:** 5
**Confidence:** 4
**Ethics Flag:** 1

**Summary:**

The paper presents a dataset called REM (Reasoning over Embodied Multi-frame trajectories) and multimodal large language models' shortcoming on spatial relationships, temporal ordering, and numerical tracking across egocentric images and trajectories. For this purpose, they have made three types of dataset: Baseline, Single Frame, and Full Rotation. Baseline benchmark general capacity. They have recorded different image views in a 3D simulation along with the trajectory. For the purpose of MLLMs assessment, they have utilized both multi-view images and trajectory together. Full rotation assess object permanence capability of multi-modal models. And, single frame investigate counting capability of these models. They have demonstrated the performance in the form of accuracy across five recent multi-modal large language models

**Reasons To Accept:**

1. Enough models have been evaluated.
2. Unlike others, the dataset has been captured in a 3D controlled environment

**Reasons To Reject:**

1. It would have been better if authors would provide more detailed description on how the image and question pairs have been generated at scale. It is ambiguous to me that how question and images have been generated and make sure the questions have been remained valid to the ground truth

2. Also, "left of" and "right of" are good aspect of spatial reasoning for evaluation. It is limited and simplified. Broader evaluation elaborates their weakness better and also make it more applicable to the real world situation

3. There is no human evaluation presented in the paper to provide the baseline regarding how the perfoemance between multimodal language models and human differs

---

> ### Author Response · Authors · 2025-06-02
>
> **If possible, please read the general summarized comment before your specific response!**
>
> Thank you for recognizing REM’s value as a controlled 3D environment for systematic evaluation. Addressing your concerns:
>
> **Ground truth generation at scale.**
> Our template system generates thousands of valid questions because each relies on perfect knowledge, deterministic scene data from *Blender*.
> - *Counting:* we sum unique object IDs (e.g., `red_sphere_1`, `red_sphere_2`) across all frames where those objects are unambiguously visible.
> - *Temporal ordering:* we compare first‑appearance frame indices (e.g., if `blue_cone_3` appears at frame 10 and `green_cube_7` at frame 25, the answer is “before”).
> - *Comparisons:* we use these exact counts.
> - *Left/right:* we project 3D positions to camera coordinates for every frame—questions are generated only when ordering is consistent across **all** co‑visible frames.
>
> This scales perfectly because ground truth comes from Blender’s scene graph, not human annotation. We will document this pipeline with examples in the appendix. We consider this a strength of REM, inspired by CLEVR’s usage of Blender scene graphs.
>
> **Spatial reasoning breadth.**
> REM tests four distinct aspects beyond left/right:
> 1. *Counting across viewpoints* probes object permanence and quantity representation.
> 2. *Numerical comparison* evaluates relative quantity representation.
> 3. *Temporal ordering* assesses trajectory‑based memory.
> 4. The “left‑of” / “right‑of” directional questions evaluate spatial relations.
>
> These tasks collectively probe different facets of embodied spatial cognition identified in cognitive‑science literature. We also show in our “Rotation” experiment that models fundamentally fail at object permanence in a simplified scenario.
>
> **Human evaluation.**
> As noted in our general response, we agree and will add human baselines showing we expect near‑perfect performance given our controlled setup, validating that the tasks are solvable with proper spatial reasoning. Note that humans will need to navigate freely between images (like MLLMs accessing their context) for larger object counts.
>
> Your feedback helps us clarify REM’s comprehensive and rigorous approach to evaluating spatial reasoning!

---

> > ### Author Response · Authors · 2025-06-09
> >
> > In addition to our previous comment, we’ve begun conducting a preliminary human baseline study (n=10 trajectories, 150 questions) with initial results showing humans achieve 95.0% overall accuracy while Gemini-1.5-Flash achieves 66% overall on the same subset.
> >
> > Per question type:
> > - **Numerical counting** (Human: 95.0%, Flash: 57.5%)
> > - **Temporal ordering** (Human: 100%, Flash: 81.1%)
> > - **Comparison** (Human: 87.2%, Flash: 61.5%)
> >
> > These substantial performance gaps suggest that our frame sequences provide sufficient context for accurate reasoning and that REM captures meaningful differences between human and MLLM embodied spatial reasoning capabilities.
> >
> > We appreciate reviewers highlighting the importance of human baselines — these results help contextualize model performance and demonstrate the benchmark’s diagnostic value. We will add complete results to Table 2 and include comprehensive methodology details in the appendix, including human dataset construction, interface design, and participant instructions.

---

### Official Review · Reviewer_iEfL · 2025-05-14

**Rating:** 7
**Confidence:** 3
**Ethics Flag:** 1

**Summary:**

This paper constructs a synthetic 3D dataset consisting of simple 3D shapes and the camera moves along a trajectory. Different views along the trajectory are saved, and a whole trajectory consisting multi-view images are used to evaluate MLLMs. They evaluated a few mainstream MLLMs and found their limitations.

=== Update after author response ===

I appreciate the authors put a lot of effort to create this dataset. While with a few limitations, this dataset still provides certain use to evaluate the 3D spatial perception and spatial memory of MLLMs. Therefore I'd like to raise the rating to accept.

**Questions To Authors:**

N/A

**Reasons To Accept:**

1. It's kind of interesting to see how MLLMs perform under such challenging scenarios.
2. This dataset may provide motivations for future MLLMs to incorporate long term memory and trajectory-wise consistent object representations.

**Reasons To Reject:**

1. There are no object movements. Only the camera moves. This is a limitation in terms of simulating real world scenarios.
2. The scene is not truly 3D, as all objects are at the same level, although with different heights. It may be more appropriate to be called "2.5D". One can argue that in many scenarios such as the street, the scenes are "2.5D" as well, however, making the scene multiple layers along the height would make the dataset more challenging and map to some real-world use cases. The authors should point out this limitation, and could view it as a future work.
3. When too many frames are input to GPT-4o or other proprietary models, the results may not make much sense, because these models are not specifically optimized to take inputs of so many images.

---

> ### Author Response · Authors · 2025-06-02
>
> Thank you for recognizing REM's value in revealing MLLM limitations and motivating trajectory‑consistent representations.
>
> **Object movement.**
> Our focus on camera movement only is a deliberate methodological choice. By keeping objects static, we isolate pure spatial reasoning and viewpoint integration from motion‑tracking complexity—establishing a critical baseline before adding object dynamics. We will clarify this design rationale and note dynamic objects as a valuable future extension.
>
> **2.5D scenes.**
> You correctly identify our scenes as 2.5D. This design enables systematic study of spatial‑reasoning limitations relevant to many real scenarios (e.g., street navigation, indoor robotics). While we will acknowledge this more explicitly and note full 3D as a future direction, the current design already reveals severe model limitations even in simplified settings.
>
> **Frame‑count scaling.**
> Your intuition about frame limits is validated by our results—performance significantly degrades with trajectory length (Figures 4‑7). This finding is a key contribution, quantifying exactly how current models fail to scale to longer visual sequences and motivating architectural improvements.
>
> We appreciate how your feedback helps us better articulate these methodological choices and their diagnostic value. *If you have time, please also read the general comment for more context.*

---

> > ### Comment · Reviewer_iEfL · 2025-06-06
> > **Thanks for your response**
> >
> > I appreciate the authors put a lot of effort to create this dataset. While with a few limitations, this dataset still provides certain use to evaluate the 3D spatial perception and spatial memory of MLLMs. Therefore I've updated my rating from 6 to 7.

---

### Author Response · Authors · 2025-06-02

**We sincerely appreciate the reviewers' insightful feedback, which will help strengthen REM.**
Below we address the three main concerns:

1. **Ground‑truth robustness.**
   Our ground truth leverages Blender’s perfect scene knowledge—each object has unique IDs, visibility is computed via pixel counting, and spatial relationships use camera‑space coordinates. Questions are generated **only** when unambiguous (e.g., directional questions require consistent left‑right ordering across *all* visible frames). Our verification uses strict keyword matching that flags ambiguous responses as incorrect. We will add detailed documentation with visual diagrams to the appendix; ground‑truth robustness is one of REM’s greatest strengths.

2. **Human evaluation.**
   Because the tasks seemed simple (a human might just take notes), we initially relied on spot‑checking rather than a rigorous human study—an oversight now being rectified. Since receiving the reviews, we have begun a human evaluation using a frame‑navigation interface that accommodates working‑memory constraints while enabling fair comparison with models. We aim to include preliminary results before the discussion ends; a full evaluation will be added in the final version (Tables 2 & 3, with graphs in the appendix).

3. **Controlled‑environment value.**
   Inspired by CLEVR, our simplified objects (8 colors × 3 shapes) intentionally isolate spatial reasoning from perception. This design reveals fundamental limitations—for example, even with perfect visual discriminability, GPT‑4o catastrophically fails (80–89 % error) to recognize that similar configurations at 0° and 180° contain different objects. Natural scenes would blur whether failures stem from perception or reasoning; our controlled setup pinpoints the core deficit.

We will expand the appendix with comprehensive methodology details, including prompting strategies plus the QA generation and verification pipelines. Together with the forthcoming human baseline, these additions will further enhance REM’s contribution.

---

### Decision · Program_Chairs · 2025-07-08

**Decision:**

Accept

**Comment:**

Four reviewers provided diverging reviews for this benchmark paper. Concerns revolve mainly around the realism of the stimuli and the lack of detail about the methodology of dataset construction and of the evaluation of VLMs on the benchmark. That said, the AC tends to agree that while with faults, the suggested benchmark has its merits for testing spatial relations and could be of use to the community. The authors are encouraged to include the details of the discussion with reviewers in their final manuscript.